# Natural Killer Cells in Graves’ Disease: Increased Frequency but Impaired Degranulation Ability Compared to Healthy Controls

**DOI:** 10.3390/ijms26030977

**Published:** 2025-01-24

**Authors:** Daniela Gallo, Eliana Piantanida, Raffaella Bombelli, Silvia Lepanto, Antonino Bruno, Matteo Gallazzi, Giorgia Bilato, Marina Borgese, Denisa Baci, Lorenzo Mortara, Maria Laura Tanda

**Affiliations:** 1Endocrine Unit, Department of Medicine and Surgery, University of Insubria, ASST Dei Sette Laghi, 21100 Varese, Italy; eliana.piantanida@uninsubria.it (E.P.); silvia.lepanto@asst-settelaghi.it (S.L.); maria.tanda@uninsubria.it (M.L.T.); 2Immunology and General Pathology Laboratory, Department of Biotechnology and Life Sciences, University of Insubria, 21100 Varese, Italy; raffaella.bombelli@uninsubria.it (R.B.); antonino.bruno@uninsubria.it (A.B.); gbilato@uninsubria.it (G.B.); denisa.baci@gmail.com (D.B.); 3Laboratory of Innate Immunity, Unit of Molecular Pathology, Biochemistry, and Immunology, Istituto di Ricovero e Cura a Carattere Scientifico MultiMedica, 20138 Milan, Italy; 91matteogallazzi@gmail.com; 4Department of Medicine and Technological Innovation, University of Insubria, 21100 Varese, Italy; marina.borgese@uninsubria.it; 5Laboratory of Molecular Cardiology Laboratory, IRCCS-Policlinico San Donato, 20097 Milan, Italy

**Keywords:** Graves’ disease, flow cytometry, natural killer cells, pathogenesis, innate immunity

## Abstract

Graves’ disease (GD) is an autoimmune disorder, driven by the appearance of circulating autoantibodies (Ab) against the thyroid stimulating hormone (TSH) receptor, thus causing hyperthyroidism. While antithyroid drugs, the only available treatment for GD, carry a significant risk of relapse, advances in immunology could pave the way for more effective therapies. Natural killer (NK) cells, divided into cytotoxic CD56^dim^ and cytokine-secreting CD56^bright^ subsets, regulate immune responses through cytokine production and cell lysis and may play a role in the pathogenesis of GD. To investigate their involvement, we conducted flow cytometry on peripheral blood samples from 131 GD patients at various stages (disease onset, on antithyroid drugs, and in remission) and 97 age- and sex-matched healthy controls (HC). We analyzed NK cell subsets, activating (CD16, CD69, NKG2D, NKp30) and inhibitory receptors (CD161, NKG2A), degranulation (CD107a), and intracellular cytokines expression (interferon γ, tumor necrosis factor α). Statistical comparisons were made between GD patients and HC and across disease stages. GD patients had a higher frequency of total NK cells (*p* < 0.028) and CD56^bright^ NK cells (*p* < 0.01) but a lower frequency of CD56^dim^ NK cells (*p* = 0.005) compared to HC. NK cells in GD patients expressed activating receptors more frequently, except for NKG2D, but had decreased cytokine expression and degranulation ability. At GD onset, patients had higher frequencies of total NK cells, CD56^bright^ NK cells, and NK cells expressing activating receptors compared to patients receiving ATD treatment and those in remission. CD161^+^ NK cells were lower at GD onset and returned to levels of HC following treatment. Correlation analysis revealed that free thyroxine (FT4) levels were inversely correlated with CD107a^+^ NK cells (*p* < 0.05) and positively correlated with CD69^+^ NK cells (*p* < 0.01). These findings suggest that hyperthyroidism impairs NK cell degranulation, with the increased frequency of NK cells potentially compensating for their reduced function. This dysfunction may contribute to the unregulated immune response in GD, highlighting NK cells as a potential target for novel therapeutic strategies.

## 1. Introduction

Graves’ disease (GD), a common autoimmune form of hyperthyroidism, develops in susceptible subjects following exposure to various environmental and endogenous factors [1]. GD is clinically characterized by hyperthyroidism and goiter [2]. The disease might be associated with extra-thyroidal manifestations, with Graves’ orbitopathy being the most common. The hallmark of GD lies in the presence of autoantibodies (Ab) that act as agonists of the thyroid stimulating hormone (TSH) receptor (R) on thyroid cells, leading to excessive thyroid hormone production and thyroid gland enlargement. Under physiological conditions, the pituitary–thyroid axis tightly controls thyroid hormone secretion. TSH-R antibodies bypass this regulatory mechanism, directly precipitating the clinical manifestations of GD [2,3]. Available treatment modalities include antithyroid drugs (ATDs), radioactive iodine, and thyroid surgery. None of these conventional treatment strategies targets GD pathogenesis [2,4]. ATDs, which include methimazole, its precursor carbimazole, and propylthiouracil, inhibit thyroid hormone synthesis. After 15 to 18 months of ATD therapy, hyperthyroidism is generally resolved, and TSHR-Ab levels become undetectable. However, up to 50% of patients may experience recurrence within the first year following ATD withdrawal [2]. This is likely related to the negligible effect of ATDs on the autoimmune process underlying the disease. Several alternative treatments are under investigation in clinical trials, including monoclonal antibodies targeting the TSH-R (KI-70), allosteric modulators of the TSH-R response (Antag-2, Antag-3, Org 274179, VA-k-14), and peptides designed to immunize immature dendritic cells to the TSH-R (ATX-GD-59) [4,5]. Other immunomodulators under examination are monoclonal antibodies targeting B and T cell activation and survival and the recycling of immunoglobulins. Although the thyroid immune infiltrate in GD is predominantly made of lymphocytes, mechanisms responsible for GD onset and persistence extend beyond adaptive immunity. A deeper understanding of the immune processes underlying the onset and progression of GD could enable the development of more effective therapeutic strategies, potentially increasing the proportion of patients achieving definitive remission through medical treatment. Since clinical characteristics of GD patients at disease onset only partially predict the highly variable risk of relapse [6], it is plausible to hypothesize that a deeper understanding of the immune processes could enhance prediction accuracy. 

Innate immune cells, including dendritic cells, macrophages, and natural killer (NK) cells infiltrate the thyroid gland where they are directly involved in antigen presentation or modulate this critical process through cell-to-cell contact and the release of cytokine and chemokines (Figure 1). NK cells, a subset of innate lymphocytes, contribute to antiviral and antitumor responses through their cytotoxic activity, which is mediated by the release of cytotoxic granules or antibody-dependent cellular cytotoxicity (ADCC) [7,8,9]. NK cells can also preserve immunological homeostasis by regulating immune responses and detecting abnormal cells without prior sensitization [10,11,12,13,14]. The two major subsets of peripheral NK cells in humans are the cytotoxic CD56^dim^CD16^+^ (CD56^dim^) NK cell subset (90–95% of peripheral NK) and the low cytotoxic CD56^bright^CD16^-/low^ (CD56^bright^) NK subset, which has high plasticity and produces large amounts of cytokines. The cytotoxic CD56^dim^ CD16^+^ NK cell subset represents the more mature subset, while the CD56^bright^ CD16^−/low^ subset consists of immature NK cells that can rapidly expand and release various inflammatory factors, including interferon γ (IFNγ), tumor necrosis factor α (TNFα), granulocyte-macrophage colony-stimulating factor, or regulatory soluble factors (such as interleukins IL-10, IL-13), depending on the stimulatory signals or cytokines/chemokines produced in the microenvironment. These cells can become cytotoxic in response to inflammatory signals. Immature NK cells can acquire the mature CD56^dim^NK cell phenotype following maturation, with the downregulation of CD56 and the acquisition of CD16 molecule and killer cell immunoglobulin-like receptors (KIRs). During the steady state and maturation, these two NK cell subsets act as sentinel cells against tumoral, infected, or stressed cells. The fine balance of all integrated activating and inhibitory signals on invariant NK surface receptors drives NK cell activation or silencing. For example, engagement of certain KIR receptors, involved in recognizing self-molecules belonging to MHC Class I (MHC-I), inhibits NK cell effector functions. Host cells that exhibit low or altered MHC-I expression or increased surface expression of ligands for receptors that activate NK cells, such as NKG2D, CD69, or the natural cytotoxic receptor NKp30, become targets for NK cells. Previous studies suggested that an imbalanced array of stimulatory and inhibitory receptors impairs NK lytic function, reducing cancer control and, potentially, promoting autoimmune phenomena [12,13,14,15,16]. It has been postulated that CD56^bright^ NK cells can acquire a regulatory role and may participate in the elimination of healthy cells as the first step of the autoimmune process, or on the contrary, in other contexts, attenuate or inhibit this process by killing autoreactive T cells, dendritic cells, or pro-inflammatory macrophages. Studies on peripheral and thyroid-infiltrating NK cells in GD remain limited [16,17,18].

This comparative cross-sectional study investigates peripheral NK cells in various stages of GD, aiming to assess their distinct phenotypes and functions. These results pave the way for further functional and in vitro studies.

## 2. Results

### 2.1. Study Cohort

The study cohort comprised 131 subjects diagnosed with GD (79% female and 21% male, mean age of 49 ± 11 years) and 97 sex- and age-matched healthy controls (HC) (Table 1). GD cohort included 60 patients who were enrolled at the onset of the disease (GD_onset), 47 patients enrolled during the first course of ATD treatment (GD_ATD), and after at least one year following discontinuation of therapy with confirmed euthyroidism and negative TSH-RAb titer (GD_remission) (Table 2). No differences were observed regarding the average levels of free thyroxin (FT4), free triiodothyronine (FT3), TSH, and TSHR-Ab that the subjects had at disease onset, before ATDs were initiated.

### 2.2. NK Cell Characterization in the Whole Cohort of GD Patients and Healthy Controls

We assessed the frequency, phenotype, and degranulation efficiency of peripheral NK cells in the GD cohort and HC (Table 1). Overall, mean CD56^bright^ NK cell frequency was significantly higher in the GD cohort than in the HC group (*p* < 0.01). In contrast, the frequency of the CD56^dim^ NK cell subset was significantly lower (*p* = 0.005) (Table 1). NK cells expressing activating receptors were more frequent in the GD cohort than in the HC groups, except for the NKG2D receptor, which was more commonly expressed by NK cells belonging to the HC cohort. Specifically, total NK cells from the GD cohort showed a significantly higher expression of CD69 compared to those from the HC group (*p* = 0.007). CD56^bright^ NK cells expressing CD16 (*p* = 0.02) and CD69 (*p* = 0.04) and CD56^dim^ NK cells expressing CD16 (*p* = 0.012) and CD69 (*p* = 0.013) were more frequent in the GD cohort compared to the HC group. No differences emerged when comparing the expression of NKp30 between patients and HC. The percentage of total NK cells and CD56^bright^ NK cells expressing the inhibitory receptor CD161 was lower in the GD cohort than in the HC group (*p* < 0.01). NK cells (total NK cells and NK subsets) had a significantly reduced degranulation efficiency against K562 cells compared to HC groups (*p* < 0.001 for comparison of total NK cells, CD56^bright^ NK cells, and CD56^dim^ NK cells) (Table 1). Overall, serum FT4 levels were significantly and directly correlated with the frequencies of total NK cells (r = 0.38, *p* < 0.01), CD56^bright^ NK cells (r = 0.27, *p* < 0.05), and 69^+^ NK cells (r = 0.47, *p* < 0.01), and inversely correlated with CD56^dim^ NK cells (r = 0.227, *p* < 0.05), CD161+ CD56^dim^ NK cells (r = −0.291, *p* < 0.05), and CD107a^+^ NK cells (r = −0.289, *p* < 0.05). TSH levels were directly correlated with CD107a frequency of expression (total NK cells r = 0.266, *p* < 0.05, CD56^dim^ NK cells r = 0.353, *p* < 0.01).

### 2.3. Frequency and Phenotype of NK Cells in Patients at GD Onset Compared to HC and Other GD Subgroups

At GD diagnosis (GD_onset group), the mean frequencies of total NK cells and CD56^bright^ NK cells were higher compared to HC (respectively, *p* = 0.011 and *p* < 0.001), GD patients undergoing ATD treatment (*p* < 0.05 for comparison between total NK cells frequencies), and those in remission (respectively, *p* = 0.04 and *p* = 0.01) (Figure 2 and Figure 3). The frequency of CD56^dim^ NK cells showed an opposite trend, being significantly lower in the GD_onset group compared to HC (*p* < 0.001), GD patients undergoing ATDs (*p* < 0.01), and those in remission (*p* = 0.008). The activating receptor CD69 was expressed from a significantly higher percentage of NK cells at GD diagnosis compared to HC (respectively, *p* < 0.01 for total NK cells and CD56^bright^ NK cells; *p* = 0.04 for CD56^dim^ NK cells), GD patients undergoing ATD treatment (*p* < 0.01 for CD56^bright^ NK cells, *p* = 0.04 for CD56^dim^ NK cells), and GD patients in remission (*p* = 0.04 for CD56^bright^ NK cells), while NKG2D followed an opposite pattern of expression (Figure 3). The inhibitory receptor CD161 was notably reduced in the GD_onset group compared to the HC group (total NK cells *p* = 0.04) (Figure 3).

### 2.4. Frequency and Phenotype of NK Cells in GD Patients During Antithyroid Drug Treatment and in Remission Compared to HC

Statistical analysis did not reveal significant differences in the mean frequencies of NK cells (both total NK cells and NK subsets) comparing GD patients undergoing ATD treatment with those in remission or between these groups and HC (Figure 2 and Figure 3). Patients in remission exhibited a significantly greater frequency of total NK cells and CD56^dim^ NK cells expressing CD16 compared to HC (*p* < 0.05) (Figure 2 and Figure 3). NKG2D^+^ expression was significantly lower in the GD_remission group compared to HC (*p* < 0.05 for total NK cells) (Figure 3). Similarly to GD_onset, the frequency of NK cells expressing the inhibitory receptor CD161 was notably reduced in GD patients taking ATDs compared to HC (total NK cells, *p* = 0. 035), but in the remission group, it returned to levels comparable to HC (Figure 2 and Figure 3). The frequency of expression of NKG2A was higher during ATD treatment than in GD_onset and GD_remission, exceeding levels of HC for the CD56^bright^ NK cells subset (*p* < 0.05).

### 2.5. Degranulation Efficiency in GD Subgroups Compared to HC

NK cell degranulation efficiency was calculated as the difference in the expression of CD107a at baseline and after exposure to a common target of cytotoxicity, as described in the “Materials and Methods” section. Degranulation efficiency was significantly reduced in the GD group compared to HC (Table 1), with no differences among GD subgroups (Figure 4). Specifically, at GD onset, serum FT4 levels were inversely correlated with the change in the frequency of CD107a^+^ NKCD56^bright^cells (*p* < 0.05). When comparing degranulation efficiency in GD_onset subgroup with HC data, the following results (mean ± SEM) were obtained: total CD107a^+^ NK cells, 27.8% ± 2.5% vs. 47.3% ± 3.3%, *p* < 0.001; CD56^dim^ CD107a+ NK cells, 28.6% ± 3% vs. 53% ± 5.4%, *p* < 0.001; CD56^bright^CD107a^+^ NK cells, 45.4% ± 3.1% vs. 60.4% ± 5%, *p* = 0.05) (Figure 4). Neither the ATD treatment nor its duration had a significant impact on degranulation efficiency, which was reduced in both patients who had received treatment for 6 months at the time of enrollment (*p* < 0.01) and those who received treatment for 12–15 months (*p* < 0.05) compared to HC. As shown in Figure 4, GD patients in remission had a significantly lower change in degranulating NK cells compared to HC (*p* < 0.01).

### 2.6. Frequency of NK Cells Expressing Intracellular Cytokines

Intracellular expression of TNFα and IFNγ was significantly lower (*p* < 0.05) comparing the entire GD population with HC. Intracellular expression of IFNγ was directly correlated with that of TNFα and CD107a. Compared to HC, TNFα positive NK cells were significantly lower in the GD_onset group (GD_onset versus HC: *p* < 0.05 for comparison of total NK cells, CD56^bright^ NK cells, CD56^dim^ NK cells) and GD_remission group (GD_remission vs. HC: *p* < 0.01 for total NK cells and *p* < 0.01 for comparison of CD56^bright^ NK cells and CD56^dim^ NK cells (Figure 5). IFNγ positive NK cells were lower in all GD subgroups compared to HC, and this was particularly evident when comparing GD patients in remission to HC (*p* < 0.05 for total NK cells, CD56^bright^ NK cells, and CD56^dim^ NK cells). The GD_remission group had significantly lower CD56^bright^ NK cells expressing TNFα and IFNγ compared to GD patients during ATD treatment (*p* < 0.05) (Figure 5).

## 3. Discussion

We detected abnormalities in blood frequencies, phenotypes, and functions of NK cells in GD patients. Although the implications are still to be elucidated, these results further support the possible role of NK cells in the onset and progression of GD. Protecting tissues from pathogens, malignancies, and excessive immune responses is essential for overall health. Tissue-resident immune cells with memory or memory-like characteristics are increasingly recognized for their critical roles in non-lymphoid tissues. It is now evident that NK cells also contribute to tissue protection.

NK cells typically constitute 10–15% of peripheral lymphocytes. NK cells migrate and reside in numerous tissue niches where they serve as vigilant sentinels, rapidly reacting to various stimuli, including tissue damage [8,19,20,21,22]. Although NK cells do not require prior antigen exposure for activation, they can achieve a form of memory (trained immunity), which enhances their response to a subsequent similar stimulus [23,24,25]. NK cells recruited to tissues exhibit both cytolytic function and cytokine secretion, which intensify upon re-exposure to pathogens, suggesting a role in defending against pathogen re-encounter [26]. In certain conditions, long-lived tissue-resident NK cells, recruited after pathogen exposure, acquire an unexpected role in maintaining tissue health. These cells help prevent excessive immune activation, thereby protecting against tissue damage and the development of autoimmunity [27]. This protective action is achieved by exerting negative feedback on activated macrophages or immature dendritic cells and by suppressing autoreactive B or T lymphocytes. NK cell function is tightly regulated by a complex balance between inhibitory and activating signals delivered from surface receptors. Interaction between inhibitory receptors on NK cells and specific ligands on host cells such as HLA complex I molecules prevent NK cell degranulation and killing. Conversely, binding between activating receptors and target ligands induces NK cell activation. It has been argued that the CD56^bright^ NK cell subset maintains immune homeostasis, orchestrating both innate and adaptive immunity through the release of cytokines, including IFNγ, TNFα, granulocyte-macrophage colony-stimulating factor, immunoregulatory cytokines IL-5, IL-10, IL-13, and the chemokines [23,28,29,30,31,32].

To our knowledge, there are no data on NK cells in human GD tissues. This is likely because thyroid biopsy is not commonly used to diagnose GD, except in cases of cold thyroid nodules. The immune infiltrate in thyroid nodules may differ from that in the surrounding thyroid tissue. While tissue obtained from thyroid surgery—performed to treat GD—could offer valuable insight into immune infiltrates during ATD treatment, collecting tissue from GD patients with a new diagnosis is not feasible. The existing data on thyroid infiltrating NK cells are limited to thyroid cancer tissues, where flow cytometry analysis has shown an increased presence of infiltrating NK cells compared to healthy thyroid tissue. These NK cells exhibited reduced cytotoxic activity, which suggests altered functionality in pathological contexts.

In our study, impaired degranulation efficiency and cytokine expression suggest NK cell failure to modulate other immune cells’ activity and attenuate the inflammatory response. A higher percentage of circulating NK cells, particularly the CD56^bright^ NK cell subset, emerged in untreated GD patients. Notably, this subset exhibited a peculiar phenotype, characterized by increased expression of activating receptors and decreased expression of the inhibitory receptor CD161. It is plausible that the augmented frequency of total and CD56^bright^ NK cells is related to the elevated prevalence of cells expressing CD69, which is involved in the early activation of NK cells. The upregulation of CD69 at GD onset may potentially facilitate the proliferation and activation of NK cells to counteract the compromised cytotoxicity [33]. Supporting this hypothesis, FT4 levels were inversely correlated to NK cell killing ability, which, in turn, was negatively correlated with the frequencies of CD69^+^ cells. Additionally, CD56^bright^ NK cells from the GD_onset, GD_ATD, and GD_remission groups had slightly increased expression of the CD16 marker compared to HC. Although this difference was not statistically significant, it may suggest the presence or induction of a subset of CD56^bright^ NK cells in GD, with altered or dysfunctional properties. This could also imply the emergence of an intermediate NK cell state, bridging immature NK cells and CD56^dim^ cells, with the potential to perform antibody-mediated functions. However, further studies are needed to clarify these observations. In untreated GD patients, the lower frequency of cells expressing the inhibitory receptor CD161 aligns with this trend. Reduced expression of the NKG2D receptor, which has been related to impaired degranulation and cytokines release, emerged in autoimmune disorders [34] such as type 1 diabetes where it affects NK’s ability to distinguish between healthy and damaged cells [12,31,35]. In arthritis (RA), NK cells have been identified as causative agents since their activating signals (IFNγ, cytotoxicity) exceed the inhibitory mechanisms. Interestingly, Yamin et al. observed higher frequencies of peripheral NK cells in severe deformative RA patients compared to HC and patients with non-deformative RA, with a similar trend in synovial fluid, albeit at lower levels [36]. Half of NK CD56^bright^ cells in synovial fluids from both deformative and non-deformative RA were CD16^+^. However, synovial fluid NK cells of erosive deformative RA had a higher capacity to release TNFα and IFNγ upon exposure to IL-2 and IL-15 [36]. In contrast, in other autoimmune diseases, such as multiple sclerosis, NK cells have a diminished capacity to inhibit inflammation, permitting its progression [11,12]. This seems to be the case for GD as most studies, including the present one, agree on impaired degranulation ability at diagnosis [17,37]. Dysfunctional NK cells may fail to orchestrate an effective immune response, particularly losing the ability to limit the expansion of dendritic cells and monocytes. We observed that euthyroidism recovery was accompanied by a decrease in the frequencies of NK cells, particularly CD69^+^ subsets. However, this was insufficient to restore cytotoxic capabilities. This factor potentially contributes to GD relapse in case of triggering factors. Indeed, in a previous prospective trial, NK cells and CD69^+^ NK cell subset decreased during ATD treatment [18]. In vitro studies testing the effect of ATD on NK cells from GD patients might offer further explanation. It might be hypothesized that an intricate milieu of hormonal, autocrine, and paracrine signals intervene during the disease, potentially modulating NK activator and inhibitory signals [38,39,40]. A bidirectional interaction between thyroid hormones (and TSH) and the immune system has emerged in physiological and pathological conditions [38,39,40,41,42,43]. NK cells, like other immune cells, contain thyroid hormones whose levels are modulated by thyroid treatments: ATDs lower thyroid hormone levels while exogenous TSH increases them [39,44]. Immune cells can also produce TSH, which may regulate autocrine thyroid hormone synthesis like the pituitary TSH. TSH, released from both the immune system and the pituitary gland, acts as a co-stimulant for cytokines, such as IL-2, IL-12, and TNFα, enhancing NK cell proliferation. Studies demonstrate that levothyroxine treatment enhances NK cell activity and IFNγ production, but not in the case of thyrotoxicosis [44,45]. In hypothyroidism, malnourished conditions and ageing thyroid hormone treatment restored immune competence, improved NK cell cytotoxicity, and amplified cytokine responsiveness. In GD patients, levothyroxine failed to enhance NK cell activity in vitro. Accordingly, hyperthyroidism in mice impairs NK cell cytotoxicity [46]. Figure 6 illustrates the hypothesis that in our cohort of patients enrolled at disease onset, NK cell function was impaired by hyperthyroidism, resulting in the loss of their regulatory inhibitory effect on other immune cells. The main limits of the present study are the observational design and the lack of a prospective follow-up. However, GD patients and HC enrolled were matched for sex and age, thus mitigating potential enrollment bias. Additionally, the severity of hyperthyroidism and TSHR-Ab titer assessed before ATD treatment were comparable among GD cohorts, thus making these cohorts comparable. To the best of our knowledge, this represents the largest study examining the phenotype and functions of distinct subsets of NK cells, namely, CD56^bright^ NK cells and CD56^dim^ NK cells, rather than only focusing on the whole population of total NK cells. These preliminary results reinforce the importance of studying the involvement of innate immunity in GD. This will support a better knowledge of GD pathogenesis but might also pave the way for new treatment strategies shaping NK response [34]. Data on tissue-resident NK cells might provide further critical insights into immune dysregulation in GD.

## 4. Materials and Methods

### 4.1. Features of Subjects Included in the Study

Patients diagnosed with GD, ages between 18 and 75 years and capable of providing informed consent, were consecutively recruited at the Endocrine Unit (Varese) between March 2017 and December 2023, following their scheduled follow-up visit. Patients with GD were classified into the following groups based on disease status: (1) first diagnosis of GD hyperthyroidism (GD_onset), (2) first course of methimazole (ATD) treatment (GD_ATD) initiated at least 6 months before enrollment and ongoing for no more than 18 months, or (3) clinical remission (GD_remission), defined as negative TSH-RAb levels and euthyroidism at least 12 months after ATD discontinuation. At GD onset, the diagnosis was confirmed based on the established criteria, including hyperthyroidism, elevated TSHR-Ab levels, and increased thyroid volume [4]. Patients who experienced hyperthyroidism relapse after one or more previous ATD treatment courses were excluded. Other exclusion criteria encompassed age younger than 18 years or older than 75 years, pregnancy, non-autoimmune thyroid disorders, active oncological or hematologic diseases, ongoing treatments affecting the immune system (such as immunotherapy, chemotherapy, and immunomodulators) or thyroid function (such as amiodarone, IFN, and lithium), mental health disorders, and inability to provide informed consent.

Age- and sex-matched euthyroid HC, without thyroid autoimmune conditions, were consecutively enrolled among volunteers attending the Transfusion Medicine Centre in Varese and Staff. The study was approved by the Institutional Review Board Ethics Committee and conducted in accordance with the Helsinki Declaration of 1975, as revised in 2013. The patient’s treatment was not influenced by the results of immunological tests and was managed based on clinical practice and the available guidelines on GD. Written informed consent for collecting anonymous data was obtained from all study participants at enrollment. All personal information was anonymized to ensure privacy and confidentiality. The study design is described in Figure 7.

### 4.2. Study Outcomes

The study aimed to investigate the role of NK cells in GD. The primary objective was to assess whether peripheral NK cells in the entire cohort of GD patients differed from those in HC. To achieve this, the study compared the frequency of total NK cells, CD56^bright^ NK cells, CD56^dim^ NK cells, the frequency of NK cells expressing activating and inhibitory receptors, and the level of degranulation after stimulus between GD patients and the HC group. The secondary objectives included comparison across different GD patient subgroups and the HC group of frequency of total NK cells, CD56^bright^ NK cells, CD56^dim^ NK cells, the frequency of NK cells expressing activating and inhibitory receptors, and the level of degranulation after stimulus exposure involved comparing these parameters across different GD patient subgroups (particularly GD onset) and the HC group. Secondary objectives also included the assessment of intracellular cytokines (IFNγ, TNFα).

### 4.3. Flow Cytometry Analysis

#### 4.3.1. Peripheral Blood Mononuclear Cells (PBMCs) Gradient Density Isolation

Venous blood samples, obtained from all GD patients and the HC, were diluted 1:2 with phosphate-buffered saline (PBS) w/o calcium and magnesium (1×) and mixed well. The solution was slowly stratified in a half volume of Lymphosep, a standardized and high-quality gradient density solution (1.077 g/mL; Biowest, Nouaillé, France). After 25 min of spin at 478× *g* and room temperature with low deceleration, the white opalescent cell layer was formed between the plasma yellow band and the Lymphosep band; this layer was carefully removed, and the cell suspension was washed twice with PBS at 416× *g*. The cells were resuspended in the RPMI1640 complete medium (Euroclone spA, Pero, Milano, Italy). and the count was performed in 0.08% Trypan Blue solution.

#### 4.3.2. Natural Killer Cell Immunophenotyping by Flow Cytometry

Immunophenotyping for circulating NK cells was performed by multicolor flow cytometry (Becton Dickinson Biosciences FACS Canto II Instrument and FACS Aria II, San Josè, CA, USA). Total PBMCs were incubated with fluorophore-conjugated monoclonal antibodies. Flow data were analyzed by FlowJo Software v.10 (BD Bioscience, CA, USA). Total NK cells were identified as CD3^−^ (PerCP-conjugated anti-CD3, BW26456 clone, Miltenyi Biotech s.r.l., Bologna, Italy), CD56^+^ (APC-conjugated anti-CD56, NCAM, REA196 clone, Miltenyi Biotech s.r.l., Bologna, Italy) cells and then assessed as CD56^bright^ and CD56^dim^ NK cells. PE- and FITC-conjugated antibodies were used to detect the percentage of NK cells (total NK cells, NK CD56^bright^ and CD56^dim^ NK cells) expressing the inhibitory receptors CD161 (W18070C clone, Biolegend Inc., San Diego, CA, USA) and NKG2A (REA110 clone, Miltenyi Biotech s.r.l., Bologna, Italy), the activating receptors CD69 (REA824 clone, Miltenyi Biotech s.r.l., Bologna, Italy), NKGD2 (CD314, REA797 clone, Miltenyi Biotech s.r.l., Bologna, Italy), NK cell protein 30 (NKp30, REA823 clone, Miltenyi Biotech s.r.l., Bologna, Italy), and the Fcγ receptor III (CD16, B73.1 clone, Biolegend Inc., CA, USA) (the gating strategy is shown in Appendix A).

#### 4.3.3. Degranulation Assay

PBMCs from GD patients and HC were co-cultured with the human immortalized myelogenous leukemia line K562 cells (1:1 ratio), incubated with FITC-conjugated anti-CD107a mAb (REA792 clone, Miltenyi Biotech s.r.l., Bologna, Italy) for 4 h (37 °C, 5% CO_2_). Degranulation of total NK cells, CD56^bright^, and CD56^dim^ NK cell subsets was obtained by flow cytometry detection of CD107a production; flow data were analyzed using the FlowJo v10 software. Basal levels of degranulation were subtracted from NK cells/K562 co-culture levels to determine NK cell degranulation efficiency; data on CD107a frequency of expression referred to this delta (the gating strategy is shown in Appendix A). Several reports indicate that CD107a, also known as lysosomal-associated membrane protein-1 (−1), is a good marker of NK cell degranulation and activation, and CD107a expression correlates with NK cell-mediated lysis of target cells. Instead, the degranulation capacity of NK cells does not overlap entirely with the cytokine production capacity because different cytokines could be released, which is why intracellular staining or ELISA of cytokines of interest is requested. K562 cell line, an undifferentiated human erythroleukemic MHC-I negative cell line, is a remarkable NK cell target, and it is a bigger physiological stimulus for NK cells than stimulation with phorbol-12-myristate-13-acetate (PMA) and ionomycin [46,47,48]. The choice of this cell target is used by many research groups in the study of innate immune responses against tumors [49] and autoimmune diseases since it efficiently detects small changes in the cytolytic function of NK cells [25,33,50].

#### 4.3.4. Intracellular Staining for Cytokine Detection

To assess the synthesis of cytokines, PBMCs from GD patients and HC were cultured in complete RPMI1640 in a 24-well plate at 5% CO_2_ and 37 °C for 24 h at 1 × 10^6^ cells/mL and 2 × 10^6^ cells/well. The PBMCs were stimulated for 1 h with 10 ng/mL of phorbol-12 myristate -13 acetate (PMA, Merck KgaA, Darmstadt, Germany) and 500 ng/mL of ionomycin (Merck KgaA, Germany). The protein transport inhibitors Golgi StopTM and Golgi PlugTM (BD Bioscience, San Diego, CA, USA) were added (4 μL every 6 ml of cell culture at 2 × 10^6^/mL) to increase intracellular cytokine accumulation. After 3 h, cells were washed, counted, and split in FAC tubes. The PBMCs were stained with CD56^-^APC and CD3-PerCP antibodies at 4 °C for 30 min in the dark, except for the blank cytometry control. After washing with PBS (phosphate buffer saline), the PBMCs were treated with 200 μL of Cytofix/Cytoperm™ Fixation/Permeabilization Kit (BD Bioscience, San Diego, CA, USA) at 4 °C for 30 min. The new washing with 1/10 water diluted Perm Wash (BD Bioscience, CA, USA) was performed and the cells were stained with intracellular cytokines mAb anti-IFNγ-PE (4 SB3 clone, BD Bioscience, CA, USA) and anti-TNFα-PE (Mab11, Biolegend Inc., San Diego, CA, USA) at 4 °C for 30 min in the dark (the gating strategy is shown in Appendix A).

### 4.4. Biochemistry

Serum TSH (reference range: 0.27–4.2 mU/L), FT4 (9.3–17 pg/mL) and FT3 (FT3, 2–4.4 pg/mL) concentrations were measured by electrochemiluminescence immunoassay (Analytical Unit for immunochemistry Cobas e801, Roche), TSHR-Ab title (normal value: < 1.5 U/L; upper limit of detection: 40 U/L) by second-generation radioreceptor assay (Thermofisher, Darmstadt, Germany).

### 4.5. Statistical Analyses

Statistical analyses were performed using SPSS (IBM software) version 29. Continuous data were reported as mean ± standard error of the mean (SEM). Discrete variables were reported as percentages. We report the *p*-values for cross-sectional comparisons in NK cell frequencies between groups obtained with the ANOVA test or *t*-test. Pearson correlation test was used to study correlations between several continuous variables. Graphs were made with GraphPad 5.

## 5. Conclusions

Immunological pathways contributing to the clinical course of GD remain poorly understood. The TSHR-Ab titer, directly correlated with autoreactive B lymphocytes, is a reliable and specific marker of disease activity [3]. However, other immunological factors are likely involved in initiating and sustaining the inflammatory process. This study provides important insights into the role of NK cells in the pathogenesis and clinical course of GD. Our findings suggest that NK cells, particularly the CD56^bright^ NK subset, exhibit altered phenotypes and functional impairments in GD patients. At disease onset, an increased frequency of NK cells expressing activation markers, such as CD69, was observed, alongside a decreased expression of inhibitory receptors like CD161. However, despite this heightened activation state, NK cell degranulation capacity and cytokine production were significantly reduced, indicating compromised immune regulation. These dysfunctions persisted even during ATD treatment and remission, underscoring the complex interplay between thyroid hormone levels, immune responses, and NK cell functions in GD. Moreover, the restoration of euthyroidism did not fully recover NK cell cytotoxicity, which may contribute to the high recurrence rate of GD following ATD withdrawal. These findings highlight the importance of considering NK cells, particularly in the context of their activation and regulatory mechanisms, in understanding GD pathogenesis. Further research into NK cell modulation may offer novel therapeutic avenues for improving disease management and preventing relapses in GD patients.

## Figures and Tables

**Figure 1 ijms-26-00977-f001:**
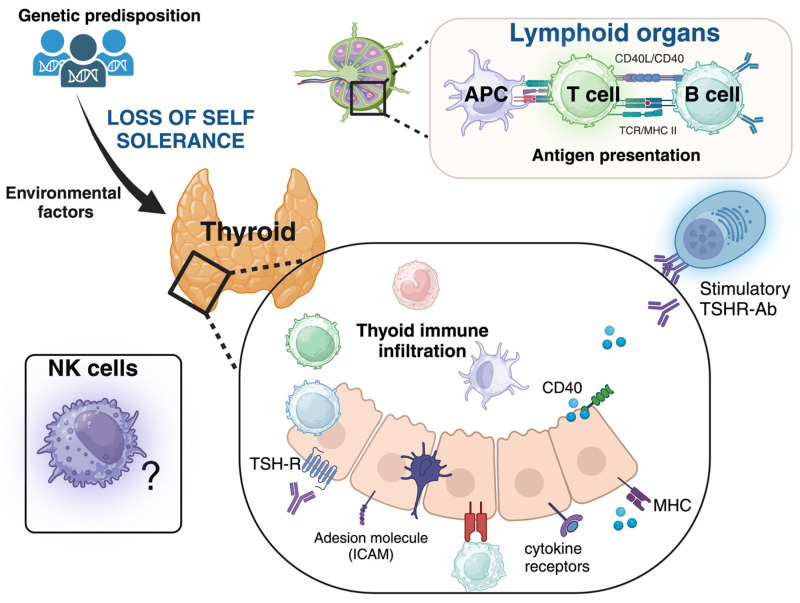
Schematic representation of Graves’ disease pathogenesis. Ab, antibody. APC, antigen-presenting cells; CD40, a cluster of differentiation 40 (a co-stimulatory protein on antigen-presenting cells), which interacts with CD40 ligand; ICAM, intercellular adhesion molecule 1; IFN, interferon; MHC, major histocompatibility complex; NK cells, natural killer cells; TNF, tumor necrosis factor; TSHR, TSH receptor. The thyroid gland presents a rich infiltrate of autoreactive T and B lymphocytes against the TSH-R, which has escaped thymic and peripheral deletion, and innate immune cells. These cells, in addition to presenting the antigen (antigen-presenting cells, APCs), contribute to the inflammatory process by secreting cytokines such as interleukins, tumor necrosis factor α, interferon γ, and chemokines. The activation of naïve T lymphocytes depends on the recognition of the MHC/peptide antigen complex on APCs and the interaction between CD40 expressed by APCs and thyrocytes with the CD154 (CD40L) ligand on activated T lymphocytes. TSH receptor-stimulating antibodies (TSHR-Ab) are the ultimate effectors in the development of GD. Created in BioRender. Gallo, D. (2024) https://BioRender.com/c64h063.

**Figure 2 ijms-26-00977-f002:**
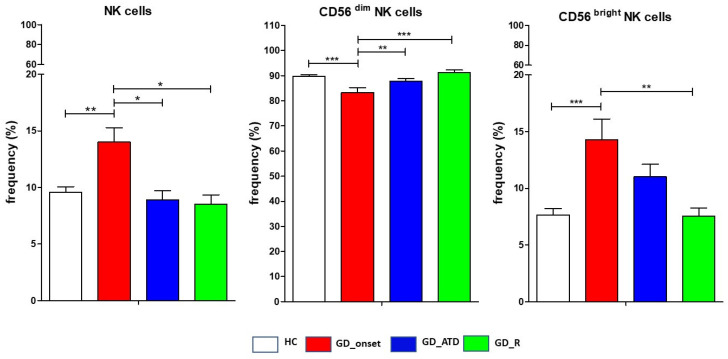
Distribution of natural killer cell subsets. Distribution of peripheral natural killer cells (total, CD56^bright^, and CD56^dim^ subsets) from Graves’ disease patients at baseline (GD_onset), during antithyroid drugs (GD_ATD), in remission (GD_remission), and healthy controls. Data are shown as mean SEM (standard error of the mean), ANOVA, * *p* < 0.05, ** *p* < 0.01, *** *p* < 0.001. The control group consisted of 97 healthy donors.

**Figure 3 ijms-26-00977-f003:**
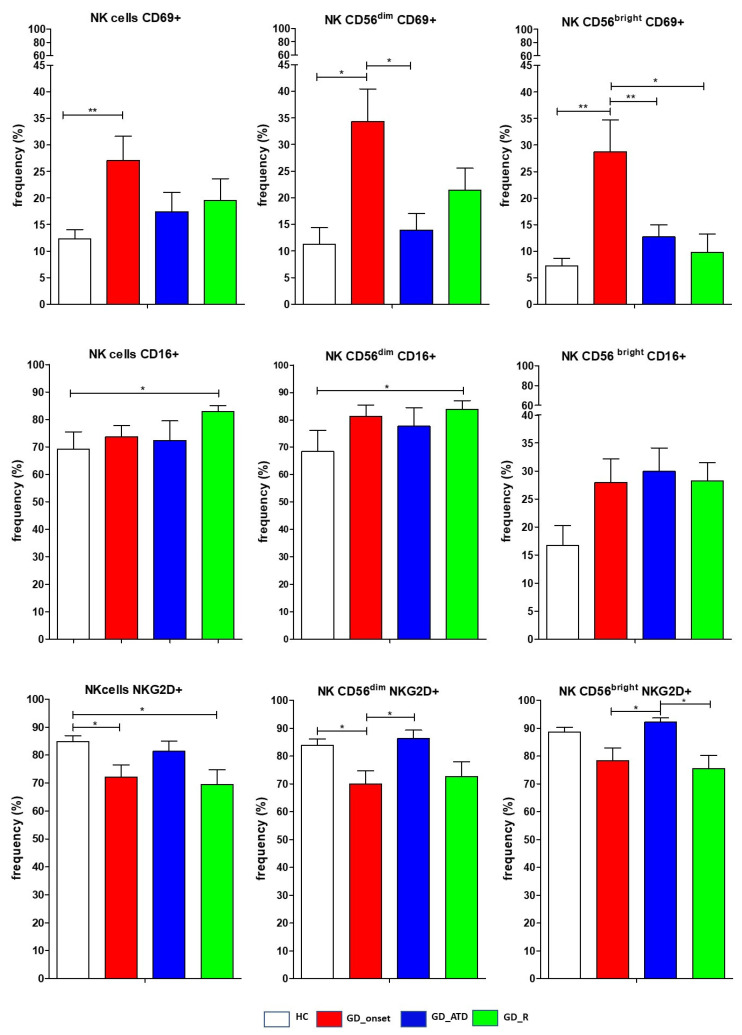
Distribution of natural killer cells expressing activating and inhibitory receptors. Histograms showed the percentages of natural killer (NK) cells (total NK cells and CD56^bright^ and CD56^dim^ subsets) expressing activating (CD69, CD16, NKp30, NKG2D) and inhibitory (CD161, NKG2A) receptors. Data are shown as mean SEM (standard error of the mean). Statistical analysis was performed using the ANOVA test, * *p* < 0.05, ** *p* < 0.01. HC, healthy controls; GD, Graves’ disease patients; ATD, methimazole; R, remission. GD patients and HC were consecutively enrolled. Forty age- and sex-matched healthy donors served as the control group. A total of 32 GD patients had data for NKp30 and NKG2A, 55 for CD16, 80 patients for CD161, 98 for NKG2D, and 103 for CD69.

**Figure 4 ijms-26-00977-f004:**
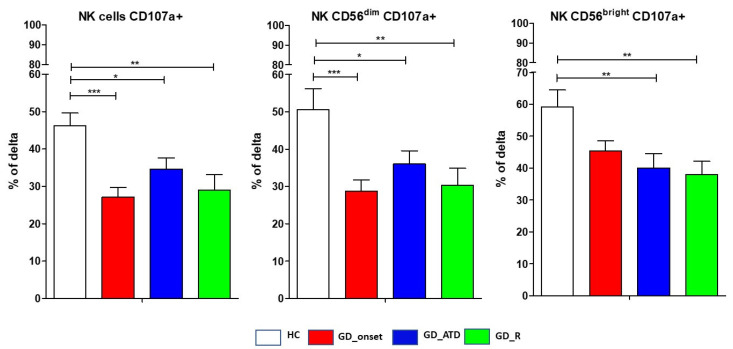
Distribution of CD107a expressing natural killer cells. Histograms showed changes in the percentages of peripheral natural killer (NK) cells expressing CD107a after stimulation compared to baseline. Data are shown as mean SEM (standard error of the mean); ANOVA test, * *p* < 0.05, ** *p* < 0.01, *** *p* < 0.001. HC, healthy controls; GD, Graves’ disease patients; ATD, methimazole; R, remission. GD patients and healthy controls were consecutively enrolled. A total of 35 sex-matched healthy controls and 92 GD patients had data for CD107a. Data are presented as mean ± standard error of the mean (SEM).

**Figure 5 ijms-26-00977-f005:**
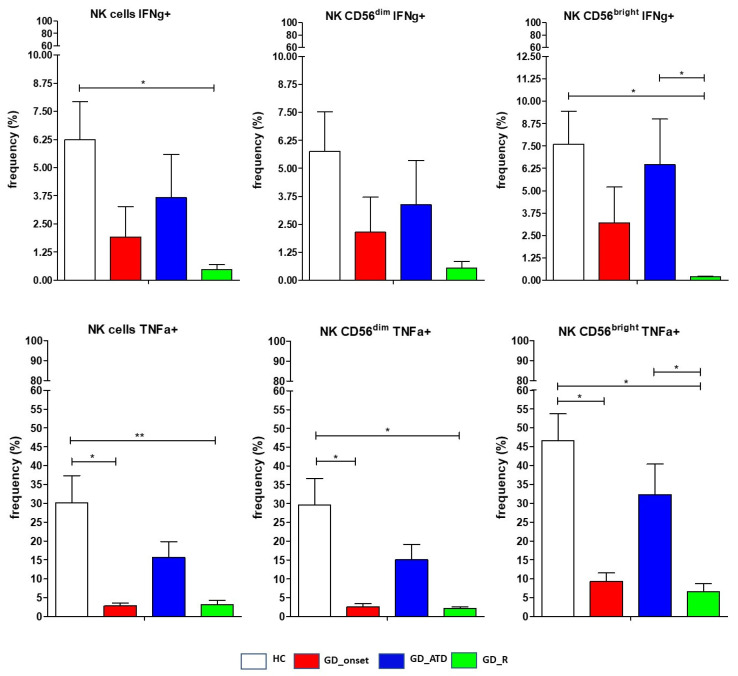
Distribution of natural killer cells expressing IFNγ and TNFα. Data are shown as mean SEM (standard error of the mean); ANOVA test, * *p* < 0.05, ** *p* < 0.01. HC, healthy controls; GD, Graves’ disease patients; ATD, methimazole; R, remission. GD patients and healthy controls were consecutively enrolled. Ten age- and sex-matched healthy donors served as the control group. Twenty-nine GD patients equally distributed among GD subgroups had data for IFNγ and TNFα.

**Figure 6 ijms-26-00977-f006:**
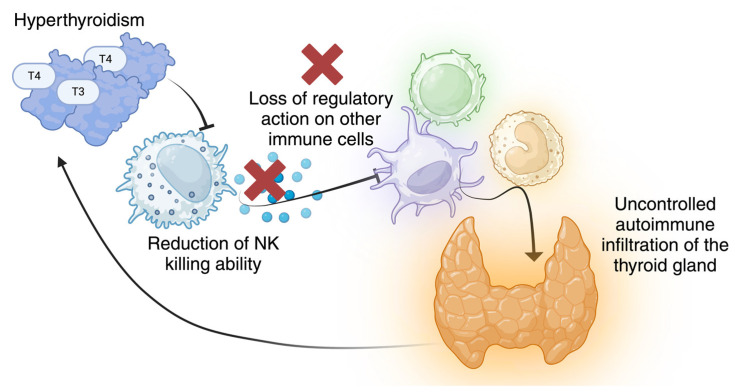
A cartoon illustrating the potential implications of natural killer (NK) cells in Graves’ disease pathogenesis. T4, thyroxine; T3, triiodothyronine. Created in BioRender. Gallo, D. (2024). https://BioRender.com/b61w450.

**Figure 7 ijms-26-00977-f007:**
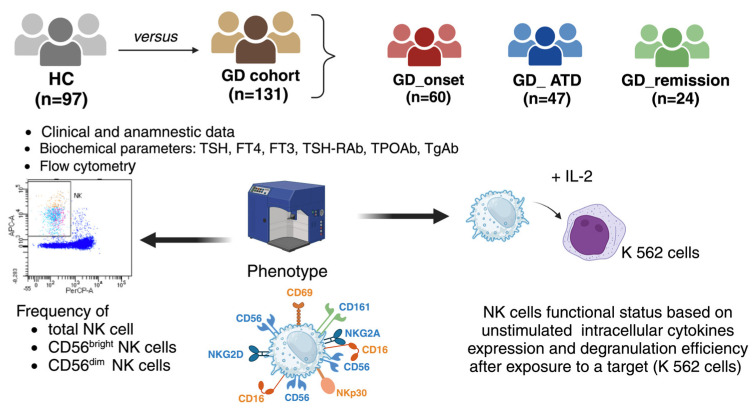
Schematic representation of the design of the study. GD, Graves’ disease; HC, healthy controls; NK, natural killer cells; Tg-Ab, thyroglobulin antibodies; TPO-Ab, thyroid peroxidase antibodies; TSHR-Ab, TSH receptor antibodies. Created in BioRender. Gallo, D. (2024). https://BioRender.com/a90z099.

**Table 1 ijms-26-00977-t001:** Demographic features and flow cytometry results for natural killer (NK) cells in the whole Graves’ disease cohort and healthy controls.

	GD Cohort (n = 131)	HC (n = 93)
*Demographic feauture*		
Age, Years	47 ± 11	41 ± 11
Sex, %F	79	77
*NK cells (frequency)*		
NK tot	11.7 ± 0.7	9.5 ± 0.9
**NK CD56^dim^**	**86.5 ± 1.01**	**88.6 ± 1.1**
**NK CD56^bright^**	**11.8 ± 0.93**	**7.7 ± 0.9**
*NK cells expressing CD69 (frequency, %)*		
**NK tot**	**23.8 ± 2.7**	**12.3 ± 1.2**
**NK CD56^dim^**	**26.3 ± 3.2**	**11.3 ± 2.3**
**NK CD56^bright^**	**18.8 ± 3**	**7.2 ± 1.3**
*NK cells expressing CD16 (frequency, %)*		
NK tot	77.4 ± 2.5	69.1 ± 5.8
**NK CD56^dim^**	**82.5 ± 2.4**	**68.4 ± 7**
**NK CD56^bright^**	**29 ± 2.2**	**16.7 ± 3.2**
*NK cells expressing NKp30 (frequency, %)*		
NK tot	63.0 ± 4.1	67.4 ± 5.8
NK CD56^dim^	61.9 ± 4.2	64.7 ± 6.2
NK CD56^bright^	68.1 ± 3.5	68.3 ± 5.2
*NK cells expressing NKG2D (frequency, %)*		
NK tot	74.6 ± 2.5	84.8 ± 2.5
NK CD56^dim^	75.9 ± 2.7	83.9 ± 3.2
NK CD56^bright^	81.9 ± 2.4	88.6 ± 3.6
*NK cells expressing CD161 (frequency, %)*		
NK tot	77.4 ± 2.7	79.3 ± 3
NK CD56^dim^	81.5 ± 2.9	75.0 ± 3
**NK CD56^bright^**	**28.59 ± 2.9**	**60.8 ± 3.9**
*NK cells expressing NKG2A (frequency, %)*		
NK tot	41.9 ± 4.3	46.5 ± 4.8
NK CD56^dim^	38.2 ± 4.3	42.6 ± 4.5
**NK CD56^bright^**	**84.7 ± 2.8**	**69.1 ± 7**
*NK cells expressing CD107a (Delta frequency, %)*		
**NK tot**	**29.9 ± 1.8**	**47.3 ± 3**
**NK CD56^dim^**	**31.2 ± 2.4**	**52.85 ± 5**
**NK CD56^bright^**	**41.6 ± 2.1**	**60.4 ± 5**

**Bold data** identified significant differences (*p* < 0.05) comparing the whole Graves’ disease cohort (GD) and healthy controls (HC). Ninety-seven age- and sex-matched healthy controls serve as a control group for NK cell distribution (total NK, CD56^bright^ NK cells, and CD56^dim^ NK cells); forty healthy donors had data for the following markers: NKp30, NKG2A, CD16, CD161, NKG2D, and CD69. All GD patients had data for NK cell distribution; 103 subjects had data for CD69, 98 for NKG2D, 80 patients for CD16, 55 for CD16, and 32 for NKp30 and NKG2A. NK tot, frequency of peripheral total NK on blood mononuclear cells; CD56^bright^, frequency of peripheral CD56^bright^ NK cells on NK cells; CD56^dim^, frequency of peripheral CD56^dim^ NK cells on NK cells. Data are presented as mean ± standard error of the mean (SEM).

**Table 2 ijms-26-00977-t002:** Demographic and biochemical features of Graves’ disease patients.

	GD_Onset	GD_ATD	GD_Remission
Number of patients	60	47	24
Body mass index (Kg/m^2^)	22 ± 4	24 ± 5	27 ± 6
Age (years)	45.8 ± 11	47.1 ± 10	54 ± 14
Active smokers (number)	22	24	4
FT4 (pg/mL)	38.5 ± 20	12 ± 4	9.7 ± 1.6
FT3 (pg/mL)	12 ± 8	3.4 ± 1	3 ± 0.5
TSH (mcU/mL)	0.02 ± 0	2.6 ± 2	1.8 ± 0.9
TSHR-Ab (U/L)	14.7 ± 11	12 ± 10	1.4 ± 1

Data (means ± SD or number were collected at the time of enrollment. ATD, antithyroid drugs; GD, Graves’ disease; FT4, free thyroxine; FT3, free triiodothyronine; TSH, thyroid-stimulating hormone; TSHR-Ab, thyroid-stimulating hormone autoantibodies. Data are presented as mean ± standard deviation (SD).

## Data Availability

Data are available upon motivated request to D.G. and L.M.

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
