# Peer review of "Natural Killer Cells in Graves’ Disease: Increased Frequency but Impaired Degranulation Ability Compared to Healthy Controls"

_ijms, 2025, doi:10.3390/ijms26030977_

Round 1
Reviewer 1 Report
Comments and Suggestions for Authors
Daniela Gallo and colleagues have provided a well-characterized analysis of the role of specific subsets of natural killer (NK) cells in Graves' disease. Remarkable is the large amount of patients examined, data collected and the amount of work performed. This study not only highlights the involvement of NK cells in the disease's pathogenesis but also suggests future research into new biomarker analyses. While the study is well-written and effectively presented, the following suggestions aim to improve its clarity and comprehensiveness:
- Abstract: Please revise the abstract for greater clarity. Clearly outline the research objectives, the nature of Graves' disease, and the potential applications of your findings. Additionally, remove section headers such as "Methods," "Results," etc., from the abstract to enhance readability.
- Introduction: Expand on the characteristics of the NK cell subsets discussed in your study to provide a more detailed context for their role in Graves' disease.
- Figures:
- Increase the font size of labels, error bars, and statistical annotations across all figures.
- Ensure consistent alignment of titles and graphs.
- Standardize the formatting of all graphs (e.g., in Figure 2, NK cell axis labels are bold, but others are not).
- Address readability issues, especially in Figure 4. Specify the Y-axis, % of delta CD107a on the total NK?
- Discussion: correct the term "simulation" in the third line to "stimulation."
- K562 Cells: Provide a clear explanation of why K562 cells were used in your study and how their use impacts the percentage of CD107a+ cells. Include details on your strategy and discuss this in the results section for better coherence.
- M&M:
- Include the isolation methods used for PBMCs.
- Provide the clone numbers for all antibodies used in the study.
- Cytokine Detection: Clarify how IFN-γ and TNF levels were detected, as these are secreted cytokines. Specify that you measured intracellular levels, and include this information in both the M&M and results sections.
- Supplementary Figure: Add a supplementary figure illustrating all gating strategies used in the study. This will help readers better understand the flow cytometry data analysis.
Author Response
Dear Editor and Reviewers,
We appreciate the extensive and supportive efforts of the Editors and the two Reviewers, who have raised important concerns, and enriching comments to our manuscript. Their comments and criticisms have helped us to improve our manuscript, and we thank you all very much indeed. In the revised version of the manuscript, we have highlighted all changes using yellow color.
Reviewer Comments
Reviewer 1
The Authors are very grateful to Reviewer 1 for her/his balanced and constructive criticisms, which helped us to improve our manuscript.
General comments:
Comment 1. Daniela Gallo and colleagues have provided a well-characterized analysis of the role of specific subsets of natural killer (NK) cells in Graves' disease. Remarkable is the large amount of patients examined, data collected and the amount of work performed. This study not only highlights the involvement of NK cells in the disease's pathogenesis but also suggests future research into new biomarker analyses.
Response: We thank Reviewer 1 for her/his positive comment.
Specific comments:
Comment 2. Abstract: Please revise the abstract for greater clarity. Clearly outline the research objectives, the nature of Graves' disease, and the potential applications of your findings. Additionally, remove section headers such as "Methods," "Results," etc., from the abstract to enhance readability.
Response: We thank Reviewer 1 for her/his criticism, which guided us in improving the abstract. We have extensively revised the abstract for better clarity. As suggested, we improved the description of research objectives, focusing on Graves’ disease nature, we better described results and their potential implication for Graves’ disease treatment.
“Graves' disease (GD) is an autoimmune disorder, driven by the appearance of circulating autoantibodies (Ab) against the thyroid stimulating hormone (TSH) receptor, THUS causing hyperthyroidism. While antithyroid drugs, the only available treatment for GD, carry a significant risk of relapse, advances in immunology could pave the way for more effective therapies. Natural Killer (NK) cells, divided into cytotoxic CD56dim and cytokine-secreting CD56bright subsets, regulate immune responses through cytokine production and cell lysis, and may play a role in the pathogenesis of GD. To investigate their involvement, we conducted flow cytometry on peripheral blood samples from 131 GD patients at various stages (disease onset, on antithyroid drugs, and in remission) and 97 age- and sex-matched healthy controls (HC). We analyzed NK cell subsets, activating (CD16, CD69, NKG2D, NKp30) and inhibitory receptors (CD161, NKG2A), degranulation (CD107a), and intracellular cytokines expression (interferon γ, tumor necrosis factor α). Statistical comparisons were made between GD patients and HC, and across disease stages. GD patients had a higher frequency of total NK cells (p < 0.028) and CD56bright NK cells (p < 0.01), but a lower frequency of CD56dim NK cells (p = 0.005) compared to HC. NK cells in GD patients more frequently expressed activating receptors, except for NKG2D, but had decreased cytokine expression and degranulation ability. At GD onset, patients had higher frequencies of total NK cells, CD56bright NK cells and NK cells expressing activating receptors compared to patients receiving ATD treatment and those in remission. CD161+NK cells were lower at GD onset and returned to levels of HC following treatment. Correlation analysis revealed that free thyroxine (FT4) levels were inversely correlated with CD107a+ NK cells (p < 0.05) and positively correlated with CD69+ NK cells (p < 0.01). These findings suggest that hyperthyroidism impairs NK cell degranulation, with the increased frequency of NK cells potentially compensating for their reduced function. This dysfunction may contribute to the unregulated immune response in GD, highlighting NK cells as a potential target for novel therapeutic strategies” (page 1 of the revised version of the manuscript).
Comment 3. Introduction: Expand on the characteristics of the NK cell subsets discussed in your study to provide a more detailed context for their role in Graves' disease.
Response: We thank the Reviewer 1 for the excellent suggestions. In response, the introduction has been thoroughly revised. We have expanded the description of NK cells (page 3 of the revised manuscript):
“The two major subsets of peripheral NK cells in humans are the cytotoxic CD56dimCD16+ NK cell subset (90-95% of peripheral NK) and the low cytotoxic CD56brightCD16-/low subset, which has high plasticity and produces large amounts of cytokines. The cytotoxic CD56dimCD16+NK cell subset represents the more mature subset, while the CD56bright CD16-/low subset consists of immature NK cells that can rapidly expand and release various inflammatory factors including interferon g (IFNg), tumor necrosis factor a (TNFa), granulocyte-macrophage colony-stimulating factor or regulatory soluble factors (such as interleukins IL-10, IL-13), depending on the stimulatory signals or cytokines/chemokines produced in the microenvironment. These cells can become cytotoxic in response to inflammatory signals. Immature NK cells can acquire the mature CD56dimCD16+ NK cell phenotype, following maturation, with the downregulation of CD56, and the acquisition of CD16 molecule and killer cell immunoglobulin-like receptors (KIRs). During the steady state and maturation, these two NK cell subsets act as sentinel cells against tumoral, infected or stressed cells. The fine balance of all integrated activating and inhibitory signals on invariant NK surface receptors drives NK cell activation or silencing. For example, engagement of certain KIR receptors, involved in recognizing self-molecules belonging to MHC class I (MHC-I), inhibits NK cell effector functions. Host cells that exhibit low or altered MHC-I expression, or increased surface expression of ligands for receptors that activate NK cells, such as NKG2D, CD69, or the natural cytotoxic receptor NKp30, become targets for NK cells. Previous studies suggested that an imbalanced array of stimulatory and inhibitory receptors impairs NK lytic function, reducing cancer control and potentially promoting autoimmune phenomena (Errore. L'origine riferimento non è stata trovata.-Errore. L'origine riferimento non è stata trovata.). It has been postulated that CD56bright NK cells can acquire a regulatory role and may participate in the elimination of healthy cells as the first step of the autoimmune process, or on the contrary, in other contexts, attenuate or inhibit this process by killing of autoreactive T cells, dendritic cells or pro-inflammatory macrophages.” (Page 3 of the revised version of the manuscript).
We have also provided a more detailed overview of Graves' disease, including its clinical features and management (page 2 of the revised manuscript). This emphasizes the importance of a deeper understanding of the disease's pathogenesis and therapeutic approaches.
“Graves’ disease (GD), a common autoimmune form of hyperthyroidism, develops in susceptible subjects following exposure to various environmental and endogenous factors (1). GD is clinically characterized by hyperthyroidism, and increased thyroid volume GOITER (2). The disease might be associated with extra-thyroidal manifestations, with Graves’ orbitopathy being the most common. The hallmark of GD lies in the presence of autoantibodies (Ab) that act as agonists of the thyroid stimulating hormone (TSH) receptor (R) on thyroid cells, leading to excessive thyroid hormone production and thyroid gland enlargement. Under physiological conditions, the pituitary-thyroid axis tightly controls thyroid hormone secretion. TSH-R antibodies bypass this regulatory mechanism, directly precipitating the clinical manifestations of GD (2, 3). Available treatment modalities include antithyroid drugs (ATDs), radioactive iodine, and thyroid surgery. None of these conventional treatment strategies targets GD pathogenesis (2, 4). ATDs, which include methimazole, its precursor carbimazole, and propylthiouracil, inhibit thyroid hormone synthesis. After 15 to 18 months of ATD therapy, hyperthyroidism is generally resolved, and TSHR-Ab levels become undetectable. However, up to 50% of patients may experience recurrence within the first year following ATD cessation (2). This is likely related to the negligible effect of ATDs on the autoimmune process underlying the disease. Several alternative treatments are under investigation in clinical trials, including monoclonal antibodies targeting the TSH-R (KI-70), allosteric modulators of the TSH-R response (Antag-2, Antag-3, Org 274179, VA-k-14), and peptides designed to immunize immature dendritic cells to the TSH-R (ATX-GD-59) (4). Other immunomodulators under examination are monoclonal antibodies targeting B and T cell activation and survival and the recycling of immunoglobulins. Although the thyroid immune infiltrate in GD is predominantly made of lymphocytes, mechanisms responsible for GD onset and persistence extend beyond adaptive immunity. A deeper understanding of the immune processes underlying the onset and progression of GD could enable the development of more effective therapeutic strategies, potentially increasing the proportion of patients achieving definitive remission through medical treatment. Since clinical characteristics of GD patients at disease onset only partially predict the highly variable risk of relapse (6), it is plausible to hypothesize that a deeper understanding of the immune processes could enhance prediction accuracy.” (Page 2 of the revised version of the manuscript)
Comment 4. Figures:
Increase the font size of labels, error bars, and statistical annotations across all figures.
We thank the Reviewer for the suggestion. We apologize for the low quality of the figures, which were improved.
Comment 5. Figures: Ensure consistent alignment of titles and graphs.
We thank the Reviewer for the criticism. We apologize if the figures were not aligned with titles and legends, and we worked to better organize the alignment of the text and figures.
Comment 6. Figures: Standardize the formatting of all graphs (e.g., in Figure 2, NK cell axis labels are bold, but others are not).
We apologize for these problems in the graphs. We revised all graphs to ensure standardization.
Comment 7. Figures: Address readability issues, especially in Figure 4. Specify the Y-axis, % of delta CD107a on the total NK?
Response: We agree with the Reviewer’s criticism and we revised all graphs, especially in Figure 4 to ensure better readability and nomenclature.
Comment 6. Discussion: correct the term "simulation" in the third line to "stimulation.
Response: Misspelling corrected.
Comment 7. K562 Cells: Provide a clear explanation of why K562 cells were used in your study and how their use impacts the percentage of CD107a+ cells. Include details on your strategy and discuss this in the results section for better coherence.
Response: This point has been clarified, as requested, in the following sentence: “Several reports indicate that CD107a, also known as lysosomal-associated membrane protein-1 (LAMP-1), is a good marker of NK cell degranulation and activation, and CD107a expression correlates with NK cell-mediated lysis of target cells. Instead, the degranulation capacity of NK cells does not overlap entirely with the cytokine production capacity, also because different cytokines could be released, which is why intracellular staining or ELISA of cytokines of interest is requested. K562 cell line, an undifferentiated human erythroleukemic MHC-I negative cell line, is a remarkable NK cell target and it's a more physiological stimulus for NK cells than stimulation with phorbol-12-myristate-13-acetate (PMA) and ionomycin (46, 48). The choice of this cell target is used by many research Groups in the study of innate immune responses against tumors (49) and autoimmune diseases since it efficiently detects small changes in the cytolytic function of NK cells (50, 51).” (page 15 of the revised version of the manuscript). (not written in the test: the % of control values CD107a positive NK cells without K562 stimulation were about 5-15%).
Comment 8. M&M: Include the isolation methods used for PBMCs.
Response: Isolation methods used for PBMCs have been added in the Material and Methods section
“Peripheral blood mononuclear cells (PBMC) gradient density isolation. Venous blood samples, obtained from all GD patients and the HC, were diluted 1:2 with phosphate-buffered saline (PBS) w/o calcium and magnesium (1 x), mixing well. The solution was slowly stratified in a half volume of Lymphosep, a standardized and high-quality gradient density solution (1.077 g/ml; Biowest, Nouaillé, France). After 25 minutes of spin at 478 x g and room temperature with low deceleration, the white opalescent cell layer was formed between the plasma yellow band and the Lymphosep band; this layer was carefully removed, and the cell suspension was washed twice with PBS at 416xg. The cells were resuspended in RPMI1640 complete medium (Euroclone spA, Pero, Milano, Italy) and the count was performed in 0.08% Trypan Blue solution” (page 14, point 4.3 of the revised version of the manuscript)
Comment 7. M&M: Provide the clone numbers for all antibodies used in the study
Response: Thank you for this appropriate suggestion. Clone numbers have been added as you requested.
Comment 8. M&M. Cytokine Detection: Clarify how IFN-γ and TNF levels were detected, as these are secreted cytokines. Specify that you measured intracellular levels, and include this information in both the M&M and results sections.
Response: This is a well-taken point. Accordingly, we have clarified this in the result section (page 10 of the revised version of the manuscript):
“Frequency of NK cells expressing intracellular cytokines. Intracellular expression of TNFα and IFNγ was significantly lower (p < 0.05) comparing the entire GD population with HC (data not shown). Intracellular expression of IFNγ was directly correlated with that of TNFα and CD107a.”
This point has also been clarified in the material and methods section (page 15, paragraph 4.3 of the revised version of the manuscript):
“Intracellular staining for cytokines detection. To assess the synthesis of cytokines, PBMC from GD patients and HC were cultured in complete RPMI1640 in a 24-well plate at 5% CO2 and 37°C for 24h at 1*10+6 cells/ml and 2*10+6 cells/well. The PBMC were stimulated for 1h with 10 ng/ml of phorbol-12 myristate -13 acetate (PMA, Merck KgaA, Germany) and 500 ng/ml of ionomycin (Merck KgaA, Germany). The protein transport inhibitors Golgi StopTM and Golgi PlugTM (BD Bioscience, Ca, USA) were added (4μl every 6ml of cell culture at 1*10+6/ml) to increase intracellular cytokines accumulation. After 3 hours, cells were washed, counted and split in FACs tubes. The PBMC were stained with CD56-APC and CD3-PerCP antibodies at 4°C for 30 minutes in the dark except for the blank cytometry control. After washing with PBS (phosphate buffer saline), the PBMC were treated with 200μl of Cytofix/Cytoperm™ Fixation/ Permeabilization Kit (BD Bioscience, CA, USA) at 4°C for 30 minutes. The new washing with 1/10 water diluted Perm Wash (BD Bioscience, CA, USA) was performed and the cells were stained with intracellular cytokines mAb anti-IFNγ-PE (4SB3 clone, BD Bioscience, CA, USA) and anti-TNFα-PE (Mab11 Biolegend inc, CA, USA) at 4°C for 30 minutes in the dark”.
Comment 9. Supplementary Figure: Add a supplementary figure illustrating all gating strategies used in the study. This will help readers better understand the flow cytometry data analysis
Response: We thank the Reviewer for his/her suggestion. We have reported all gating strategies used in the study as supplementary materials, Suppl. Figure 1, Suppl. Figure 2 and Suppl. Figure 3 (see in the text at Supplementary material the corresponding doi).
Varese, Italy
January 17 2024
Lorenzo Mortara, PhD
Associate Professor
Immunology and General Pathology Laboratory (DBSV)
School of Medicine, University of Insubria
71, Monte Generoso street - 21100 Varese, Italy
Tel. +39.0332.39.76.10
E.mail: lorenzo.mortara@uninsubria.it
Reviewer 2 Report
Comments and Suggestions for Authors
This study investigates the altered frequency and functionality of NK cell subsets in Graves’ disease (GD). It reveals increased activation markers but impaired degranulation efficiency, particularly in CD56bright NK cells. The findings suggest NK cell dysfunction contributes to GD pathogenesis and highlights the influence of thyroid hormones on immune regulation. The manuscript is well organized, but the novelty is the major concern here, since the author published a similar paper in another journal 5 years before. Here are my concerns and suggestions.
Majors:
1. Figure Quality and Localization of Autoimmune Response. The resolution of Figure 1 is inadequate for proper evaluation, necessitating higher quality for better comprehension. Additionally, the authors should clarify the anatomical site of the autoimmune activity (thyroid), which is only implicitly addressed. A more detailed timeline and sequence of immunological events in GD, possibly in an enhanced figure, would aid understanding of the disease’s progression.
2. The abstract mentions a “double p-value” for CD56bright NK cells (“p < 0.028, p < 0.01”), which is unclear. It raises concerns about redundancy or misinterpretation. Authors should specify whether these represent different statistical tests or subsets and ensure clarity to avoid confusion.
3. The elevated expression of CD16 in CD56bright NK cells in GD raises the intriguing possibility of altered antibody-dependent cellular cytotoxicity (ADCC) activity. The authors could explore this in future studies or at least hypothesize how this finding might influence GD pathogenesis, particularly given its relevance to innate immunity and antibody-mediated mechanisms.
4. The discussion linking thyroid hormones (T3/T4) to NK cell function is intriguing but requires expansion. The inverse relationship between FT4 levels and NK degranulation efficiency (CD107a) suggests thyroid hormones modulate NK cell functionality. However, direct mechanistic exploration or reference to relevant studies is lacking. The authors should delve deeper into hormonal signaling pathways affecting NK cells and their implications in GD.
5. The study's reliance on peripheral blood NK cells is acknowledged as a limitation. Adding a discussion on tissue-resident NK cells within the thyroid and their potential roles in GD could enrich the narrative. Tissue-specific phenotypes and functionalities might provide critical insights into localized immune dysregulation in GD.
The manuscript provides valuable insights into NK cell dynamics in GD but needs refinements in data presentation, statistical clarity, and causal interpretations. Addressing these issues will significantly enhance the study’s impact and reliability.
Comments on the Quality of English LanguageThe English could be improved to more clearly express the research.
Author Response
Dear Editor and Reviewers,
We appreciate the extensive and supportive efforts of the Editors and the two Reviewers, who have raised important concerns, and enriching comments to our manuscript. Their comments and criticisms have helped us to improve our manuscript, and we thank you all very much indeed. In the revised version of the manuscript, we have highlighted all changes using yellow color.
Reviewer Comments
Reviewer 2
The Authors are very grateful to Reviewer 2 for her/his balanced and constructive criticisms, which helped us to improve our manuscript.
General comments:
Comment 1. This study investigates the altered frequency and functionality of NK cell subsets in Graves’ disease (GD). It reveals increased activation markers but impaired degranulation efficiency, particularly in CD56bright NK cells. The findings suggest NK cell dysfunction contributes to GD pathogenesis and highlights the influence of thyroid hormones on immune regulation. The manuscript is well organized, but the novelty is the major concern here, since the author published a similar paper in another journal 5 years before. Here are my concerns and suggestions.
Response: We thank the Reviewer for her/his thoughtful comments. We are grateful for appreciating the manuscript structure. Our research group has been investigating the role of immune cells, particularly natural killer (NK) cells, in thyroid autoimmune disorders. While we have published previous studies on this topic, this manuscript is the first to comprehensively analyze NK cell subsets in a large cohort of patients with Graves' disease. Our study spans not only the disease onset and early phases of antithyroid drug treatment but also the remission phase. The data presented here have not been published elsewhere. It seems that the Reviewer may have referenced the manuscript "Gallo D, Piantanida E, Gallazzi M, et al. Immunological Drivers in Graves' Disease: NK Cells as a Master Switcher. Front Endocrinology (Lausanne). 2020;11:406." However, it is important to note that this is a review article, not an original research study, and as such, it does not overlap with the novel data presented in our current work.
Majors:
Comment 2. Figure Quality and Localization of Autoimmune Response. The resolution of Figure 1 is inadequate for proper evaluation, necessitating higher quality for better comprehension. Additionally, the authors should clarify the anatomical site of the autoimmune activity (thyroid), which is only implicitly addressed. A more detailed timeline and sequence of immunological events in GD, possibly in an enhanced figure, would aid understanding of the disease’s progression.
Response: We thank the Reviewer for her/his criticisms. We tried to improve Figure 1, which has been completely modified. The legend has been expanded to support a better understanding.
(see new modified Figure 1 in the article).
Ab, antibody. APC, Antigen-presenting cells; CD40, cluster of differentiation 40 (a co-stimulatory protein on antigen-presenting cells) which interacts with CD40 ligand; ICAM, intercellular adhesion molecule 1; IFN, interferon; MHC, Major Histocompatibility Complex; NK cells, Natural Killer cells; TNF, tumor necrosis factor; TSHR, TSH receptor. The thyroid gland presents a rich infiltrate of autoreactive T and B lymphocytes against the TSH receptor (TSH-R), which have escaped thymic and peripheral deletion, and innate immune cells. These cells, in addition to presenting the antigen (antigen-presenting cells, APCs), contribute to the inflammatory process by secreting cytokines such as interleukins, Tumor Necrosis Factor α, Interferon γ, and chemokines. The activation of naïve T lymphocytes depends on the recognition of the MHC/peptide antigen complex on APCs and the interaction between CD40 expressed by APCs and thyrocytes with the CD154 (CD40L) ligand on activated T lymphocytes. TSH receptor-stimulating antibodies (TSHR-Ab) are the ultimate effectors in the development of GD. Created in BioRender. Gallo, D. (2024) https://BioRender.com/c64h063.
(page 2 and 3 of the revised version of the manuscript).
Comment 3. The abstract mentions a “double p-value” for CD56bright NK cells (“p < 0.028, p < 0.01”), which is unclear. It raises concerns about redundancy or misinterpretation. Authors should specify whether these represent different statistical tests or subsets and ensure clarity to avoid confusion.
Response: We thank the Reviewer for her/his comment. Since Reviewer 1 also raised concerns regarding the abstract, we have thoroughly revised it. To address the specific point raised, we would like to clarify that in the previous version, the first p-value (p < 0.028) referred to a comparison with total NK cells, while the second p-value (p < 0.01) pertained to CD56^bright NK cells. We have explicitly addressed this distinction in the revised manuscript, which now reads: “GD patients had a higher frequency of total NK cells (p < 0.028) and CD56^bright NK cells (p < 0.01), but a lower frequency of CD56^dim NK cells (p = 0.005) compared to HC.” (page 1 of the revised manuscript)”.
Comment 4. The elevated expression of CD16 in CD56bright NK cells in GD raises the intriguing possibility of altered antibody-dependent cellular cytotoxicity (ADCC) activity. The authors could explore this in future studies or at least hypothesize how this finding might influence GD pathogenesis, particularly given its relevance to innate immunity and antibody-mediated mechanisms.
Response: We greatly appreciate this valuable suggestion. In response, we have expanded the characterization of the two NK cell subsets in the introduction. Additionally, in the discussion, we have specifically addressed the interesting point raised regarding CD16 expression on CD56^bright NK cells
“The cytotoxic CD56dimCD16+NK cell subset represents the more mature subset, while the CD56bright CD16-/low subset consists of immature NK cells that can rapidly expand and release various inflammatory factors including interferon (IFN), tumor necrosis factor (TNF), granulocyte-macrophage colony-stimulating factor or regulatory soluble factors (such as interleukins IL-10, IL-13), depending on the stimulatory signals or cytokines/chemokines produced in the microenvironment. These cells can become cytotoxic in response to inflammatory signals. Immature NK cells can acquire the mature CD56dimCD16+ NK cell phenotype, following maturation, with the downregulation of CD56, and the acquisition of CD16 molecule and killer cell immunoglobulin-like receptors (KIRs). During the steady state and maturation, these two NK cell subsets act as sentinel cells against tumoral, infected or stressed cells. The fine balance of all integrated activating and inhibitory signals on invariant NK surface receptors drives NK cell activation or silencing. For example, engagement of certain KIR receptors, involved in recognizing self-molecules belonging to MHC class I (MHC-I), inhibits NK cell effector functions. Host cells that exhibit low or altered MHC-I expression, or increased surface expression of ligands for receptors that activate NK cells, such as NKG2D, CD69, or the natural cytotoxic receptor NKp30, become targets for NK cells. Previous studies suggested that an imbalanced array of stimulatory and inhibitory receptors impairs NK lytic function, reducing cancer control and potentially promoting autoimmune phenomena (12-16). It has been postulated that CD56bright NK cells can acquire a regulatory role and may participate in the elimination of healthy cells as the first step of the autoimmune process, or on the contrary, in other contexts, attenuate or inhibit this process by killing of autoreactive T cells, dendritic cells or pro-inflammatory macrophages.” (pag 3 of the revised version of the manuscript).
“Additionally, CD56bright NK cells from the GD_onset, GD_ATD, and GD_remission groups had slightly increased expression of the CD16 marker compared to HC. Although this difference was not statistically significant, it may suggest the presence or induction of a subset of CD56bright NK cells, in GD, with altered or dysfunctional properties. This could also imply the emergence of an intermediate NK cell state, bridging immature NK cells and CD56dim cells, with the potential to perform antibody-mediated functions. However, further studies are needed to clarify these observations.” (..) “Interestingly, Yamin et al. observed higher frequencies of peripheral NK cells in severe deformative RA patients compared to HC and patients with non-deformative RA, with a similar trend in synovial fluid, albeit at lower levels (36). Half of NK CD56bright cells in synovial fluids from both deformative and non-deformative RA were CD16+. However, synovial fluid NK cells of erosive deformative RA had a higher capacity to release TNFα and IFNγ upon exposure to IL-2 and IL-15 (36).” (pag 12 of the revised version of the manuscript).
Comment 5. The discussion linking thyroid hormones (T3/T4) to NK cell function is intriguing but requires expansion. The inverse relationship between FT4 levels and NK degranulation efficiency (CD107a) suggests thyroid hormones modulate NK cell functionality. However, direct mechanistic exploration or reference to relevant studies is lacking. The authors should delve deeper into hormonal signalling pathways affecting NK cells and their implications in GD.
Response: This is a very well-taken point. We were glad to delve into this interesting bidirectional relationship between immune and endocrine systems. “A bidirectional interaction between thyroid hormones (and TSH) and the immune system has emerged in physiological and pathological conditions (38-43). NK cells, like other immune cells, contain thyroid hormones, whose levels are modulated by thyroid treatments: ATDs lower thyroid hormone levels while exogenous TSH increases them (39, 44). Immune cells can also produce TSH, which may regulate autocrine thyroid hormone synthesis like the pituitary TSH. TSH, released from both the immune system and the pituitary gland, acts as a co-stimulant for cytokines, such as IL-2, IL-12, and TNFα, enhancing NK cell proliferation. Studies demonstrate that levothyroxine treatment enhances NK cell activity and IFNγ production, but not in the case of thyrotoxicosis (44,45). In hypothyroidism, malnourished conditions and ageing thyroid hormone treatment restored immune competence, improved NK cell cytotoxicity, and amplified cytokine responsiveness. In GD patients, levothyroxine failed to enhance NK cell activity in vitro. Accordingly, hyperthyroidism in mice impairs NK cell cytotoxicity (46). Figure 6 illustrates the hypothesis that in our cohort of patients enrolled at disease onset, NK cell function was impaired by hyperthyroidism, resulting in the loss of their regulatory inhibitory effect on other immune cells.” (pag 12 of the revised version of the manuscript).
Comment 6. The study's reliance on peripheral blood NK cells is acknowledged as a limitation. Adding a discussion on tissue-resident NK cells within the thyroid and their potential roles in GD could enrich the narrative. Tissue-specific phenotypes and functionalities might provide critical insights into localized immune dysregulation in GD.
Response: Thank you for raising this important point. Unfortunately, there is limited data on resident NK cells in the thyroid gland, except in the context of thyroid cancer. This may be because thyroid biopsy is not commonly used to diagnose Graves' disease, except in cases of cold thyroid nodules on thyroid scintigraphy. The immune infiltrate in thyroid nodules may differ from that in the surrounding thyroid tissue. While tissue from thyroid surgery- performed when indicated to treat Graves' disease- could provide insight into immune infiltrates during antithyroid drug treatment, it is not feasible to obtain tissue from patients with a new diagnosis. This is because euthyroidism must be restored before surgery to minimize perisurgical complications. We have included a brief comment on this in the revised version of the manuscript (page 11 of the revised version of the manuscript):
“Protecting tissues from pathogens, malignancies, and excessive immune responses is essential for overall health. Tissue-resident immune cells with memory or memory-like characteristics are increasingly recognized for their critical roles in non-lymphoid tissues. It is now evident that NK cells also contribute to tissue protection. NK cells typically constitute 10-15% of peripheral lymphocytes. NK cells migrate and reside in numerous tissue niches, where they serve as vigilant sentinels, rapidly reacting to various stimuli, including tissue damage (8, 19-22). Although NK cells do not require prior antigen exposure for activation, they can achieve a form of memory (trained immunity), which enhances their response to a subsequent similar stimulus (23-25). NK cells recruited to tissues exhibit both cytolytic function and cytokines secretion, which intensify upon re-exposure to pathogens, suggesting a role in defending against pathogen re-encounter (26). In certain conditions, long-lived tissue-resident NK cells, recruited after pathogen exposure acquire an unexpected role in maintaining tissue health. These cells help prevent excessive immune activation, thereby protecting against tissue damage and the development of autoimmunity (27). This protective action is achieved by exerting negative feedback on activated macrophages or immature dendritic cells and by suppressing autoreactive B or T lymphocytes. NK cell function is tightly regulated by a complex balance between inhibitory and activating signals, delivered from surface receptors. Interaction between inhibitory receptors on NK cells and specific ligands on host cells such as HLA complex I molecules prevent NK cell degranulation and killing. Conversely, binding between activating receptors and target ligands induces NK cell activation. It has been argued that the CD56brightNK cell subset maintains immune homeostasis, orchestrating both innate and adaptive immunity through the release of cytokines, including IFNγ, TNFα, granulocyte-macrophage colony-stimulating factor, immunoregulatory cytokines IL-5, IL-10, IL-13, and the chemokines (23, 28-32). To our knowledge, there are no data on NK cells in human GD tissues. This is likely because thyroid biopsy is not commonly used to diagnose GD, except in cases of cold thyroid nodules. The immune infiltrate in thyroid nodules may differ from that in the surrounding thyroid tissue. While tissue obtained from thyroid surgery - performed to treat GD - could offer valuable insight into immune infiltrates during ATD treatment, collecting tissue from GD patients with a new diagnosis is not feasible. The existing data on thyroid infiltrating NK cells are limited to thyroid cancer tissues, where flow cytometry analysis has shown an increased presence of infiltrating NK cells compared to healthy thyroid tissue. These NK cells exhibited reduced cytotoxic activity, which suggests altered functionality in pathological contexts.”
Comments on the Quality of English Language. The English could be improved to more clearly express the research.
Response. The English has been revised to improve quality. The final version of the manuscript was also reviewed by a native English speaker with a degree in English literature for professional editing (see acknowledgements).
Varese, Italy
January 17 2024
Lorenzo Mortara, PhD
Associate Professor
Immunology and General Pathology Laboratory (DBSV)
School of Medicine, University of Insubria
71, Monte Generoso street - 21100 Varese, Italy
Tel. +39.0332.39.76.10
E.mail: lorenzo.mortara@uninsubria.it
Round 2
Reviewer 1 Report
Comments and Suggestions for Authors
I would like to thank the authors for the great work they have done, particularly during the review process, which has significantly enhanced the quality and clarity of their work, giving the recognition that rightfully deserve. The work is now ready for publication.